# Best Practices for Providing Patient-Centered Tele-Palliative Care to Cancer Patients

**DOI:** 10.3390/cancers15061809

**Published:** 2023-03-16

**Authors:** Grecia Lined Aldana, Onyinyechi Vanessa Evoh, Akhila Reddy

**Affiliations:** Department of Palliative, Rehabilitation, and Integrative Medicine, The University of Texas MD Anderson Cancer Center, Houston, TX 77030, USA; glaldana@mdanderson.org (G.L.A.); ovevoh@mdanderson.org (O.V.E.)

**Keywords:** telemedicine, palliative care, cancer, best practices, health care delivery

## Abstract

**Simple Summary:**

Telemedicine has become a permanent platform for delivering palliative care. Our review highlights the best practices that palliative care teams can incorporate to provide high-quality, interdisciplinary virtual care to cancer patients.

**Abstract:**

Cancer patients receiving palliative care may face significant challenges in attending outpatient appointments. Patients on controlled substances such as opioids require frequent visits and often rely on assistive devices and/or a caregiver to accompany them to these visits. In addition, pain, fatigue, and shortness of breath may magnify the challenges associated with in-person visits. The rapid adoption of telemedicine in response to the COVID-19 pandemic has proven to be highly beneficial for advanced cancer patients and caregivers. The hurried COVID-19-related implementation of telemedicine is now evolving into a permanent platform for providing palliative care. This review will focus on the best practices and recommendations to deliver high-quality, interdisciplinary tele-palliative care.

## 1. Introduction 

Telemedicine is the use of telecommunications to provide patients with health services in a virtual format and includes audio-video or telephone communication to connect with patients. Telemedicine has been utilized for decades, and the use of the telephone to assist in patient care can be seen even as far back as the Civil War [1]. Providers used telemedicine before the COVID-19 pandemic to deliver care to patients living in rural areas and to monitor the functional status and symptomatic burden between in-person visits [2,3]. The use of telemedicine grew exponentially in the wake of the COVID-19 pandemic, which necessitated health care delivery to be urgently adapted to the requirements of social distancing. There was an emphasis on providing high-quality care while minimizing exposure in patients at high risk for complications of COVID-19. Additional concerns arose explicitly in cancer patients receiving palliative care, especially given the high symptom burden among patients, the delicate nature of goals of care conversations, and the support patients and their families often need at this vulnerable time. Palliative care services worldwide played a vital role during the COVID-19 pandemic, and telemedicine allowed for continued care to patients and their families [4,5].

There is emerging literature on the benefits of telemedicine for cancer patients receiving palliative care [6,7,8,9]. One major advantage is eliminating travel requirements that often impose an immense physical burden on patients with limited mobility and high symptom burden [10]. Another benefit is minimizing the financial losses that traveling can represent, which could include lost wages for missed days of work for patients and caregivers, and associated travel costs such as gas, parking, meals, hotel stays, and childcare [11,12]. These are especially important in palliative care, given the physical and financial burdens imposed on patients with many medical appointments and commitments. Telemedicine allows clinicians to provide patients with timely interventions, decreasing emergency room visits and hospitalizations [13,14]. For example, a patient with uncontrolled pain or opioid-induced neurotoxicity may receive help from a tele-palliative care provider the same day and undergo opioid titration or rotation. These patients may otherwise need to access the emergency room to receive such timely interventions. Similarly, quick access to a counselor or psychologist may address acute emotional distress. A palliative care provider may even assess patients using telemedicine simultaneously in collaboration with an in-person home health provider for more comprehensive interdisciplinary management. Telemedicine also allows for large family meetings with the palliative care team, potentially bringing family members from different geographic locations to participate, thereby enhancing family engagement in the patient’s care [15,16]. Telemedicine could also facilitate joint visits with a primary care provider, oncologist, and palliative care physicians. These coordinated visits can be advantageous for cancer patients, especially during the transition of care or goals of care discussion. Telemedicine can help provide a “just-in-time” model of advance care planning and goals of care discussion in the inpatient setting by including multiple specialties and disciplines in family meetings on short notice [17]. This model enables clinicians to deliver a cohesive message while bringing their unique expertise and collaboration to develop a goal-concordant plan for patients and families.

Telemedicine reduces the need for missing or canceling in-person follow-up appointments among palliative care patients [18], which could decrease emergency room visits due to uncontrolled symptoms [19]. It can improve access to multidisciplinary palliative care consultations, such as chaplains, pharmacists, social workers, and counselors, which helps to enhance patient satisfaction [20]. It could even improve a patient’s prognosis since travel burden has been associated with negatively impacting cancer patients’ prognoses and quality of life [21]. A recent systematic review concluded that telemedicine using a videoconferencing system might be as effective as in-person care in cancer patients [22]. Therefore, an increase in telemedicine use for the delivery of outpatient cancer care might be warranted. Telemedicine can also be utilized to offer integrative therapies such as yoga and music therapy, which have the potential to enhance physical and mental wellness [23,24].

The significant social and global influence of telemedicine is undeniable, as it can serve as the sole means of providing medical assistance to both military personnel and civilians in times of war. More recently, there has been advocacy for recognizing telemedicine as a fundamental human right [25].

Tele-palliative care is increasingly being adopted worldwide, and many programs continue to participate in a high volume of tele-visits [26]. However, there are no structured guidelines for delivering tele-palliative care to cancer patients. Our review highlights the best practices palliative care teams can adopt when conducting tele-visits using a two-way interactive audio-video platform.

## 2. Best Practices for Telemedicine in Palliative Care

While telemedicine continues to be used broadly in palliative care, the best practices for conducting video visits are yet to be established. Patients switch between in-person and telemedicine visits depending on their convenience and caregiver’s availability, transportation, other appointments, the need for testing, and various other factors. The lack of standardization of telemedicine practices can lead to different experiences for patients and palliative care teams across multiple medical centers and may differ significantly from the care received during in-person appointments. A successful tele-palliative care visit relies on many aspects. Some aspects are not very different from a conventional in-person visit and are modified to suit the virtual platform. For in-person visits, the patients and families make appointments, receive appointment reminders, check in when they arrive, and get roomed. Nurses perform palliative care assessments, and physicians or advanced practice providers (APPs) then conduct a comprehensive visit to address all the domains of palliative care [27]. The Interdisciplinary Team (IDT) members are included as needed after the physician visit, followed by the conclusion and scheduling of a follow-up appointment [28]. A similar workflow must be followed for telemedicine visits to ensure patients receive the same care level. Introduction to the purpose and scope of the visit with palliative care IDT, building rapport, examining the patients, formulating an action plan, education, and counseling are necessary steps in providing good care. Allocating sufficient time in the visit to address any concerns and questions patients and families may have is vital. Feeling heard and understood by providers and obtaining the required help for pain and other symptoms are some of the main patient-reported experience measures in outpatient palliative care [9,29,30,31].

With the right tools and modifications, telemedicine can aspire to enhance the patient and palliative care provider experience significantly. We created a framework of best practices to meet this challenge of providing high-quality tele-palliative care, improving the experience of patients and caregivers, and helping standardize telemedicine practice in palliative care (Table 1). An extensive literature review and the almost 3-year experience providing predominantly tele-palliative care in our supportive care center helped formulate the best practice recommendations [3,4,18,32,33].

## 3. Best Practice Recommendations

### 3.1. The Pre-Visit Checklist

We recommend completing a checklist of items 1–2 days before the telemedicine visit. This can be performed by office staff or medical assistants before the visit. Following this checklist ensures that patients and their caregivers are adequately prepared for their visit. The pre-visit checklist should attempt to:Confirm the appointment and access to the internet and audio-video platform used during the visit.Check the patient or caregiver’s familiarity with the audio-video platform and offer assistance with downloading applications and registration.Ask the patient to log in 1-hour ahead of the appointment to ensure there are no challenges and provide ample time to seek assistance.Provide a contact for troubleshooting or a help desk if the connection is unsuccessful on the day of the visit. Access to a help desk is crucial, and centers should invest in a capable support system for the success of any telemedicine program.Confirm the patient’s intended location to ensure telemedicine is permitted in their geographical area. For example, many states in the U.S. do not allow telemedicine visits across state lines. The patient’s address and call back number during the visit can be helpful in emergencies or if a patient expresses suicidal or homicidal ideations.Identify if an interpreter service is required and make arrangements for an interpreter to assist in the conversations and join the telemedicine visit.Encourage the completion of assessments before the visit, such as the Edmonton Symptom Assessment System (ESAS) and other screening tools.Stress the importance of having a private space and dedicated time for the visit.Encourage the use of headphones if private space is not accessible.Encourage family participation, including those in various locations, and offer access to the help desk as needed.Walk them through the anticipated workflow that can be expected during the visit.

### 3.2. Patient Setting

Tele-palliative care allows the patient to experience personalized care in an environment that is familiar, which may add to the therapeutic relationship between provider and patient. Cox et al. highlight how patients who meet with physicians electronically in their homes perceive increased time spent, given that they can meet in a comfortable and familiar environment (their homes) [34]. Although patients should feel comfortable in their environment during the meeting, they should also be in a location and setting that allows them to give their full attention to the video visit. The providers should encourage patients to be in a well-lit room with a functioning camera on their phones or computers. They should promote safety and recommend that patients schedule their meetings far from distractions such as watching young children. Visits should be halted if safety is questioned, such as if a patient is driving during the tele-visit [35]. Providers should also encourage the patient or caregivers to move the camera or change positions to visualize the patient’s entire face and torso. Proper positioning will help ensure the patient and provider are giving the visit the attention required for a good therapeutic outcome. The same setting recommendations apply to family members not present with the patient and joining the visit separately. If the patients or families cannot participate in the tele-visit from a private location, headphones are strongly encouraged.

### 3.3. Palliative Care Provider Setting

Proper positioning is also essential for the provider for a successful video visit. Strasser et al. studied the importance of sitting vs. standing during oncology visits. They found that patients strongly preferred and perceived better care was provided by physicians who sat rather than stood during the visit [36,37]. It is recommended that the provider sit comfortably without distractions, be relaxed, face the camera to make adequate eye contact, and lean forward a little to convey attentiveness. Providers should move the patient’s image near the camera if the video platform allows, to enable easy eye contact. It is also crucial that the provider is in a quiet, well-lit room during the video visit, with a professional background or one that conveys a peaceful and relaxed atmosphere without any bright or flashy distractions. Avoid windows or bright lights in the background. The presence of other members of the team or staff within the video frame is discouraged.

### 3.4. Acquainting the Patient

First and foremost, verify that the patient can see and hear you. Introduce yourself and identify the patient using the institutional guidelines, for example, the name and date of birth. Once the patient’s and family’s identity (if present) are established, greet them warmly and go over what to expect during the visit. Establish a rapport early and assure them of privacy. The patients must have a sense of privacy and feel secure in their conversations with the provider. Acknowledge the limitations of the video visit and include that platforms allow only one person to speak at a time. Minimize interruptions and rely more on non-verbal responses such as nodding your head, raising your eyebrows to express concern, or leaning forwards rather than saying “hmm” or “yes” or “I know” while the patient is speaking, with an emphasis on webside manner [38,39].

Avoid eating or drinking during the visit and mute all potential distractions such as pagers, cell phones, and emails. Identify a backup plan early in the visit if there is a loss of internet connection or poor-quality audio. Prioritize listening over speaking and avoid long pauses. Focus on non-verbal means to convey empathy and respond to emotions. Avoid taking notes or working on the patient’s chart during the tele-visit. Periodically summarize your understanding of the patient’s discussion to ensure you both are on the same page. Use verbal responses such as “I wish.” and “I admire.” to convey empathy.

Telemedicine has limitations, and those must be clearly explained to the patient. For example, one cannot conduct a thorough physical examination that may be necessary for an accurate diagnosis [40]. It is essential to clarify the achievable goals regarding patient care at the onset of the meeting to ensure the nature of the tele-visit is understood. While the pre-visit checklist helps acquaint the patient with the nature of the visit to a certain extent, an additional moment to reacquaint the patient is recommended. Introducing the nature and goals of the visit at the onset helps remind the patient and any caregivers present that medical providers will take their concerns seriously and address them similarly to an in-person visit [41].

### 3.5. Assessments and IDT Visits

It is crucial to maintain a consistent workflow for the IDT during tele-visits and match the workflow to in-person visits [32]. At MD Anderson Cancer Center, our tele-palliative care workflow begins with the patient checking into a virtual waiting room environment where the registered nurse (RN) obtains a brief medical history, clarifies the reason for the visit, conducts a comprehensive symptom assessment using the ESAS as a guide, and updates the medication list in the chart. Assessments such as the Memorial Delirium Assessment Scale (MDAS), the Cut-down, Annoyed, Guilty, and Eye-opener questionnaire (CAGE), and the Eastern Cooperative Oncology Group Performance Status Scale (ECOG) can also be performed at this time. Subsequently, the nurse places the patient in the virtual waiting room and presents the report to the physician or the APP. The physician or APP then joins the video platform and proceeds with the rest of the tele-visit, expanding on data gathered during the RN assessment. Following the physician’s assessment, other interdisciplinary team members, such as the pharmacist, chaplain, counselor, nutritionist, social worker, or case manager, are invited to join the visit as needed. To replicate an in-person interdisciplinary palliative care visit, members of the IDT must be available to join the visit as needed. The provider may leave and join later to close the visit (see #7) [4]. This allows for real-time interventions to address patient needs and provide personalized care. Having the virtual involvement of the IDT frees the patient and the IDT from some of the constraints that time and distance may put on interdisciplinary intervention and helps ensure that patients receive evaluation and input from many team members simultaneously.

### 3.6. Physical Examination in Telemedicine

A proper physical exam can aid physicians in diagnosing and treating patients. Whether virtual or in-person, the findings of a physical exam can go a long way toward accurately diagnosing and helping patients by providing adequate pharmacological and psychological support. Moreover, advanced cancer patients indicate that examination is a highly positive aspect of their care [42]. In the context of a tele-visit, conducting such an exam becomes more complex through the camera’s lens. Overcoming the barrier to examining a patient via video may be possible by having a keen sense of observation and learning to perform a visual examination. Various assessments and screening tools can be used as part of the video visit with subtle modifications to enhance the virtual examination of a patient.

We propose a tele-visit examination (Table 2) that allows physicians to critically evaluate a palliative care patient. We suggest starting with a general assessment of the patient, looking for signs of pain such as grimacing, moaning, splinting, guarding, and cautious movements when changing position [43].

Providers should always ask themselves, "does the patient look uncomfortable?" Consider requesting the caregiver to check vital signs if possible since some patients possess blood pressure cuffs, pulse oximeters, etc.

You can evaluate the size of pupils, which could be particularly useful when there is a concern for non-medical opioid use (NMOU). If there is doubt that the patient may be using recreational drugs, look for mydriasis (benzodiazepines, stimulants, cannabis, or alcohol) or miosis (opiates) [43]. Invite the patient to move closer to the camera to glimpse the gums, oral mucosa, and throat. Check the tongue movement, especially in head and neck cancer patients and those with mucositis.

Through a video visit, a physician could also assess for respiratory distress, observing closely for the use of accessory muscles or even audible wheezing or cough. Consider evaluating respiratory effort using the Roth score: request the patient to take a deep breath followed by counting out loud from 1 to 30. If the maximal counting number is <10 or the counting time is <7 s, then the pulse oximeter reading is likely less than 95%. If the maximal counting number is <7 or the counting time is <5 s, then the pulse oximeter reading is likely <90% [44].

Observe carefully for any abdominal distention, protruding masses, or possible ascites. Palliative care patients are particularly affected by bowel problems [45], mainly related to the use of opioids, metabolic disturbances, reduced oral intake, poor performance status, tumor burden, or metastasis. Besides obtaining important information such as the timing of the last bowel movement, an abdominal exam assisted by the patient or family members can help you determine if the abdomen is soft, tender, or rigid and whether the findings need further in-person evaluation.

Observe the extremities for myoclonus, swelling, and erythema. Patients receiving opioids must be observed for signs of neurotoxicity (confusion, drowsiness, myoclonus). An MDAS can also aid in a neurological evaluation. A psychiatric evaluation by assessing mood, affect, psychomotor agitation or retardation, euphoria, anxiety, anhedonia, and suicidal ideation is essential.

### 3.7. Closing the Visit

Along with summarizing the visit and ensuring that the patient and family are agreeable to the proposed treatment plan, an opportunity must be provided for the patient to discuss any last-minute concerns. After completing the tele-visit, the physician can place pertinent orders, send electronic prescriptions to the pharmacy, order any testing, and schedule the next visit. The follow-up visit could be a tele-visit or in-person based on the patient’s needs and the provider’s assessment. Telemedicine can promote continuity of care as many patients may find it feasible to attend follow-up telemedicine visits compared to in-person visits [18]. Patients exhibiting NMOU behaviors or symptom distress that need a thorough physical exam can be scheduled for a short-term in-person follow-up appointment. Palliative care clinics offering telemedicine visits must have guidelines set forth to care for patients with NMOU, and follow-ups must be scheduled accordingly [33]. These patients may need random urine drug screens as part of their treatment plan.

Providers can also make arrangements for patients to complete advance directives such as medical power of attorney, living will, and out-of-hospital do not resuscitate documents. Many of these documents can be completed electronically with assistance from the IDT, and such services must be offered to patients who otherwise cannot come in person. Finally, the provider must assess the patient and caregiver’s comfort with the virtual format and ability to participate successfully in future telemedicine visits. If patients are uncomfortable with participating or unable to participate adequately in telemedicine visits, in-person appointments must be offered. The patient’s comfort with technology, cognitive abilities, and safety barriers, such as driving or engaging in other activities during the visit, must be considered before scheduling a follow-up visit (telemedicine vs. in-person).

## 4. Barriers to Telemedicine

One can follow all the best practices recommended above and still encounter barriers with telemedicine [46]. Despite efforts of acquainting patients with technology and walking them through the process of getting ready for tele-visits, some patients can continue to experience technical difficulties with connection, sound, and video, forgetting their log-ins, and other technological hiccups [11,47].

Furthermore, while previous studies comparing virtual and in-person physical exams have concluded that they may be equivalent, an in-person physical examination is sometimes necessary to evaluate the patient properly [48]. For example, new onset back pain would require an examination to evaluate for epidural disease and spinal cord compromise. The inability to conduct a thorough neurological examination could present a missed opportunity. Similarly, any new and severe pain should be followed by a thorough physical examination pertinent to the pain’s location to prevent missing diagnoses such as fractures or metastatic bone disease. Identifying patients who need in-person appointments and quickly accommodating them for a face-to-face visit is necessary for patient safety and avoiding litigation.

Another barrier in telemedicine is the inability to appropriately display empathy and emotions integral to any palliative and supportive care visit [38,49,50]. Another significant barrier is medical reimbursement. As a result of the pandemic, tele-visits are being reimbursed similarly to in-person visits, which allows providers and establishments to continue with a high proportion of virtual care. It is yet to be seen whether the COVID-19 concessions will be reversed in the future, which may lead to significantly lower reimbursement for tele-visits [4,51,52,53]. A lower reimbursement may result in a decline in providers that offer tele-palliative care, which could be a significant setback. Laws and regulations could also present a barrier to telemedicine in palliative care, especially for patients who require opioids. Some states require an in-state medical license or strictly prohibit prescribing opioids through telemedicine [4,54].

Another barrier is the potential lack of privacy for the patient. Some patients may be concerned that others in their house or the physician’s office can overhear their sensitive information during the tele-visit [55]. While this can be minimized by using headphones or the chat feature available with most virtual platforms, the perceived lack of privacy may continue to be a barrier to developing a good virtual therapeutic relationship [56].

## 5. Conclusions

Telemedicine is a valuable tool in providing patient-centered palliative care to cancer patients. Telemedicine offers hope to cancer patients for whom in-person visits are difficult due to timing, disability, pain, or other contributing factors. We proposed specific practice recommendations for providing high-quality tele-palliative care that can enhance the experience of patients and caregivers. The best practices presented here are not a complete list but are a starting point to create a more streamlined approach to delivering patient-centered care. Future studies must focus on compiling comprehensive, evidence-based best practices for tele-palliative care.

## Figures and Tables

**Table 1 cancers-15-01809-t001:** Best Practices for Telemedicine in Palliative Care.

Before the visit	➢ Confirm the appointment and ensure the patient has a smartphone or computer with the telemedicine platform installed. ➢ Encourage the patient to get familiar with the technology. Provide them with the information needed for troubleshooting, such as a help desk they can call in case of unsuccessful attempts to connect to the visit. ➢ Ask the patient to log in 1-hour before the tele-visit so they have ample time to seek assistance if needed. ➢ Confirm the patient’s location and contact information for emergencies. ➢ Identify if an interpreter is required and make arrangements for one to join the tele-visit if needed. ➢ Stress the importance of having a private space and dedicated time for the visit. Headphones may be necessary to ensure privacy. ➢ Encourage family participation, including those in various locations, and offer the same access as mentioned earlier to a help desk if needed. ➢ Walk them through the anticipated workflow that can be expected during the visit. ➢ Encourage the completion of assessments such as the ESAS and other screening tools.
During the visit	➢ Providers must be in a quiet, well-lit room with a professional background without any bright or flashy distractions. Avoid windows or bright lights in the background. ➢ Sit comfortably without distractions, be relaxed, and face the camera to make adequate eye contact. Providers may consider moving the patient’s image near the camera for easy eye contact. ➢ Greet the patient and any family warmly after ensuring they have a good audio and video connection to the visit. Introduce yourself. Reconfirm their location and contact information. ➢ Establish a rapport early during the visit and make them feel at ease. Outline the goals of the visit. Spend more time listening than talking. ➢ Use non-verbal responses such as nodding your head or leaning forwards rather than saying “hmm”, “yes”, or “I know” while the patient is speaking. ➢ Avoid eating or drinking during the visit and mute all potential distractions such as the pager, cell phone, and emails. ➢ Identify a backup plan if there is a loss of internet connection or poor-quality audio. ➢ Follow a consistent workflow. For example, the nurse conducts a comprehensive assessment using ESAS, MDAS, and CAGE questionnaires. The physician receives a detailed report from the nurse, joins the video platform, and proceeds with the rest of the telemedicine visit, expanding on the data gathered. Afterward, other interdisciplinary team members, such as the pharmacist, chaplain, counselors, nutritionist, social worker, or case manager, are invited to join the visit as needed. ➢ Conduct a telemedicine physical examination using keen observation skills and ask the patient and family to assist as needed (refer to Table 2). ➢ Use appropriate webside manner and respond to emotions. ➢ Review the assessment and plan with the patient and answer any questions they have. ➢ If appropriate, outline what you would like to focus on during the following visit (e.g., medication adjustments (increasing or weaning off opioids), goals of care).
After the visit	➢ Ensure prescriptions are ordered and referrals to other services are requested as needed. ➢ Reconcile medications, complete notes, and provide the patient with an electronic after-visit summary. ➢ Schedule a proper follow-up based on the need for an in-person vs. tele-visit and the patient’s preference. Patients exhibiting non-medical opioid use behavior or those experiencing technology-related barriers may need to be seen in person.

Abbreviations: ESAS—Edmonton Symptom Assessment System; MDAS—Memorial Delirium Assessment Scale; CAGE—Cut-down, Annoyed, Guilty and Eye-opener questionnaire.

**Table 2 cancers-15-01809-t002:** Physical examination of a palliative care patient during a tele-visit.

Constitutional	➢ Can ask patient or caregivers to check vital signs if available. ➢ Assess general appearance, grooming, and attentiveness. ➢ Look for signs of pain such as grimacing, moaning, guarding, and cautious movements when changing positions.
HEENT	➢ Assess the appearance of the eyelids (ptosis?), eye movement, and size of pupils. ➢ Look for mydriasis (benzodiazepines, stimulants, cannabis, or alcohol) or miosis (opiates). ➢ Check for lesions and masses. ➢ Invite the patient to move closer to the camera to glimpse the gums, oral mucosa, and throat. Check tongue movement, especially in head and neck cancer patients and those with mucositis. ➢ Assess facial symmetry and movement of the mouth.
Neck and chest	➢ Assess tracheal position, gross lymphadenopathy, jugular venous distension, and neck range of motion. ➢ Evaluate for any masses, may inspect wounds, and assess for chest wall tenderness with self-palpation or the help of a caregiver. ➢ May inspect breasts if needed in patients with post-mastectomy complications, inflammatory breast cancer, wounds, etc.
Respiratory	➢ Observe closely for the use of accessory muscles. ➢ Listen for audible wheezing, coughing, and sounding short of breath while speaking. ➢ Assess respiratory effort with Roth Score: request the patient to take a deep breath followed by counting from 1 to 30 in their native language. If the maximal counting number is <10 or the counting time is <7 s, then pulse oximetry is likely less than 95%. If the maximal counting number is <7 or the counting time is <5 s, then pulse oximetry is likely <90%.
Cardiovascular	➢ Assess for jugular venous distension and edema in the extremities. ➢ Can guide the caregivers in assessing capillary refill.
Abdomen	➢ Visual examination of the abdomen for distension, protruding masses, or ascites ➢ If present, take the help of others to palpate the abdomen and assess for tenderness, guarding, etc.
Musculoskeletal	➢ Examine the gait, posture, and range of motion of various joints. ➢ Observe if any movements elicit pain. ➢ Assess for tenderness (with self-palpation), muscle strength, and swelling. ➢ Assess for the need or the use of assistive devices such as wheelchairs, walkers, etc.
Neurologic	➢ Assess for alertness and orientation. ➢ Look for signs of opioid-induced neurotoxicity (confusion, drowsiness, hallucinations, myoclonus, etc.) ➢ Use screening tools such as MDAS to screen for delirium. ➢ Cranial nerves can be examined by close inspection. ➢ Check for gait abnormality, motor strength, and tremors.
Psychiatric	➢ Assess for flattened affect, psychomotor retardation or agitation, anxiety, inflated mood, mood lability, and speech pattern. ➢ Be attentive to recognize hallucinations or delusional thoughts. ➢ Assess for suicidal ideations.
Skin	➢ Check for pallor, cyanosis, rashes, pigmentation, lesions, petechiae, ulcers, etc.

Abbreviations: HEENT—head, eyes, ears, nose, and throat; MDAS—Memorial Delirium Assessment Scale.

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
