# Peer review of "Best Practices for Providing Patient-Centered Tele-Palliative Care to Cancer Patients"

_cancers, 2023, doi:10.3390/cancers15061809_

Round 1
Reviewer 1 Report
Thank you for the opportunity to review manuscript ID: cancers-2252086 entitled “Best Practices for Providing Patient-Centered Tele-Palliative Care to Cancer Patients”.
The authors provide an overview on tele-palliative care and the speedy development and implementation in many institutions during the pandemic. The paper focuses mainly on recommendations and best practices for high-quality and interdisciplinary tele-palliative care providing checklists for preparation, during and after a tele-visit including physical tele-examination hints.
The paper is well written and concise; it provides very practical tips, which have their important place in the clinical work environment.
Overall I find this work important, well-written and practical and I recommend it for publication.
Author Response
Thank you very much for your positive feedback.

Reviewer 2 Report
The article is interesting and concerns a very modern aspect with interesting future developments.
Although it is a conceptual and substantially descriptive article, the topic of telemedicine applied to cancer patients is of absolute importance.
In the introduction, the authors correctly pointed out the importance of telemedicine and the development it has had due to the COVID-19.
This introduction is correct even if telemedicine is now a very important way of providing care that is reflected globally at a social level.
It would be appropriate to explain the importance of telemedicine at a social and global level and not just in relation to treatments. (You can see this article: DOI: 10.1177/1357633X221103829 or others as well).
In the paragraph "Barriers to Telemedicine" the authors correctly pointed out the problems and barriers related to the use of telemedicine.
This is a relevant issue because it can generate problems of a medico-legal nature. Although it would be superfluous to insert another paragraph on this aspect, it is important to at least mention it because it is a fundamental question.
Author Response
Thank you for the positive comments. We have now added the following to the introduction and barriers section of the manuscript respectively.
"A recent systematic review concluded that telemedicine using a videoconferencing system might be as effective as in-person care in cancer patients [22]. Therefore, an increase in telemedicine use for the delivery of outpatient cancer care might be warranted. Telemedicine can also be utilized to offer integrative therapies such as yoga and music therapy, which have the potential to enhance physical and mental wellness [23,24]. The significant social and global influence of telemedicine is undeniable, as it can serve as the sole means of providing medical assistance to both military personnel and civilians in times of war. More recently, there has been advocacy for recognizing telemedicine as a fundamental human right [25]."
"Furthermore, while previous studies comparing virtual and in-person physical exams have concluded that they may be equivalent, an in-person physical examination is sometimes necessary to evaluate the patient properly [48]. For example, new onset back pain would require an examination to evaluate for epidural disease and spinal cord compromise. The inability to conduct a thorough neurological examination could present a missed opportunity. Similarly, any new and severe pain should be followed with a thorough physical examination pertinent to the pain's location to prevent missing diagnoses such as fractures or metastatic bone disease. Identifying patients who need in-person appointments and quickly accommodating them for a face-to-face visit is necessary for patient safety and avoiding litigation."
